# Ecuadorian Woods: Building Material Selection Using an Entropy-COPRAS Comparative Analysis Based on the Characterization of Ecuadorian Oak and Guayacan Timber

**DOI:** 10.3390/biomimetics9070443

**Published:** 2024-07-19

**Authors:** Juan Francisco Nicolalde, Javier Martínez-Gómez, Paúl Dávila, Johanna Medrano-Barboza, Jaime Vinicio Molina-Osejos

**Affiliations:** 1Facultad de Ciencias Técnicas, Universidad Internacional Del Ecuador UIDE, Quito 170411, Ecuador; junicolaldego@uide.edu.ec (J.F.N.); padavilaal@uide.edu.ec (P.D.); 2Universidad de Alcalá, Departamento de Teoría de la Señal y Comunicación, Área de Ingeniería Mecánica, Escuela Politécnica, 28805 Alcalá de Henares, Madrid, Spain; 3Facultad de Arquitectura e Ingenierías, Universidad Internacional SEK, Albert Einstein s/n and 5th, Quito 170302, Ecuador; johanna.medrano@uisek.edu.ec (J.M.-B.); jaime.molina@uisek.edu.ec (J.V.M.-O.)

**Keywords:** multicriteria decision methods, material characterization, mechanical resistance, thermal conductivity

## Abstract

Considering that global awareness for sustainable development has risen to face environmental damages, different building materials have been considered from a mechanical perspective. In this sense, considering the richness of South America regarding its woods, the Guayacan and the Ecuadorian oak timbers have not been previously characterized. The present research has performed mechanical, thermal, and moisture content characterizations to acknowledge the benefits of considering these materials for the building industries. In this sense, Guayacan has been shown to have lower thermal conductivity, making it ideal for thermal insulation; the oak from Manabi showed the best compressive strength; while the oak from El Oro stands with the best tensile strength; and the oak from Loja showed the best modulus of elasticity. On the other hand, all the materials were compared by multicriteria decision methods to select the best, by using the COPRAS method driven by the objective entropy-weighted method, showing that the oak from Loja is the best choice considering the advantage that presents with the modulus of elasticity. In this sense, it is concluded that regarding the mechanical properties, there is not much difference for the compression, bending, and tensile strength; nevertheless, for the modulus of elasticity the oak from Loja stands out, making it a factor to be considered in the selection of a wood for building applications that is corroborated through multicriteria decision methods.

## 1. Introduction

The growing awareness of climate change has led to research on sustainable materials, where wood, as a renewable resource, can be considered as an excellent material with excellent qualities that can become a key element of the future bioeconomy [1]. On the other hand, wood is considered a renewable resource with mitigation impacts, and waste is reused for other wood products and energy production [2]. Consequently, a progressive societal demand for sustainable, effective, and durable materials must be considered.

In this sense, in the construction sector, wood has been considered a useful material for structural beams, frames, and flooring. This status was achieved considering its excellent strength-to-weight ratios, thermal and acoustic insulation properties, and especially, its high sustainability rates and recyclability characteristics [3]. However, traditional construction materials such as timber in Ibero-American countries have not been considered as a primary material [4]. Instead, brick buildings have been used as the most common construction practice. However, in the past, wood was used for buildings but resulted in poor-performance housing due to construction techniques with energy-inefficient results [5], which leaves a door open for sustainable development research in the sector [4]. Based on the energy efficiency of the materials for building purposes, it is important to consider the design of buildings with different materials and with different architectural forms, to reduce a large part of the energy losses, given that the transfer of energy between the environment and the building generates an energy demand [6]. On the other hand, it is important to state that the use of wood as a sustainable material for buildings must come with a responsible practice that in South America has been resolved by each country’s government that has promoted the maintenance of an economically productive and sustainable forest industry [7]. In this sense, energy efficiency in buildings is an important topic to be disclosed; even more, it becomes more relevant considering that energy efficiency is considered a reliable strategy to achieve sustainable development, where developing countries are still in an improvement process compared to industrializations [8]. Taking in mind these considerations, in Ecuador there are many varieties of trees, the most commonly mobilized species are Laurel (*Laurus nobilis*), Doncel (*Zanthoxylum rhoifolium*), Zapote (*Pouteria sapota),* Balsa (*Ochroma pyramidale*), Ceibo (*Ceiba pentandra*) Chalviande (*Virola sebifera)*, Arenillo (*Andira inermis)*, Tamburo (*Vochysia leguina)*, Colorado (*Cissus striata*) and Copal (*Bursera glabrifolia*) [9] while timbers such as pigue (*Piptocoma discolor*), jacaranda (*Jacaranda mimosifolia*), huayacan (*Guaiacum officinale*), laurel mascarey (Hieronyma Alchorneoides), pachaco (*Schizolobium parahyba*), cutanga (*Parkia multijuga*), batea caspi (*Cabralea canjerana*), cedro (*Cedrus*), Ferman Sanchez (*Triplaris cumingiana*), capirona de monte (*Calycophyllum spruceanum*), manglillo (*Hedyosmum mexicanum*), tachuelo (*Zanthoxylum precerum*), peine de mono (*Apeiba membranacea*), guayacan (*Tebebuia chrysantha*), and pechiche (*Vitex cymosa*) are commercial species that in early stages are identified by farmers and promoted in on-farm agroforestry [10].

Considering Ecuador’s sustainable woods, the oak (*Nectandra montana*) is a traditional material that can be named. The term oak is used to refer to many species of trees of the genus *Quercus*, native to the Northern Hemisphere, and occasionally also to species of other genera of the same family (*Fagaceae*) or even of other families, as in the case of some South American species of Nothofagus (*fam. Nothofagaceae*) [11]. Oak wood has a straight grain, a coarse grain, and you can clearly distinguish sapwood from heartwood. Sawing is difficult, but wear and tear on saws is normal. Drying should be slow and gentle, as there is a risk of superficial flaws appearing. This timber is durable against the action of fungi, but sensitive to anobiums, lyctids, and cerambycids, and moderately durable to termites [12]. On the other hand, oak has good aptitude for obtaining flat panels if it is previously dried and does not present problems for gluing. The wood of this species of oak is used in interior carpentry, furniture, sleepers, and waterworks, and was formerly widely used in Gothic cathedrals and shipbuilding. It is also a wood that is used to make quality wine, being comparable to oaks of French or American origin [13]. On the other hand, local research has found that Guayacan wood (*Caesalpinia paraguariensis)* can be used as a material for buildings. Guayacan has been used traditionally as a source of fuelwood and charcoal, it has been used for furniture and indigenous people used it for tools. Even more, the tree can reach its maturity between 10 and 20 years, despite different climate conditions such as rainfalls or extreme temperatures, making it a solid candidate for studying its thermal properties in building applications [14].

The characterization of building materials is ruled by different standards that allow for analysis of the heat exchange of building materials. There are different methods such as the ISO 5151: 2010 [15] that reports total cooling capacity, the AISI/ASHRAE Standard 16–2016 for cooling and dehumidification [16], and the AHRI Standard 340/360:2015 for cooling and dehumidification [17], among others [18], such as the heat balance proposed in the regulations of air conditioning, refrigeration, and heating.

In this way, Guayacan is a potential material to be exploited as a construction timber, while oak has proven to be useful in traditional applications, and considering the region’s common practices brick and cement are used for buildings. However, there are different characteristics to be considered, keeping in mind that the building components must be chosen to fulfil the requirements of serviceability and safety limits [19]. In this sense, while the thermal properties of materials allow them to determine the thermal behavior regarding different temperature variations [20], the susceptibility to humidity that might arise in the structure can mean a risk of biodegradability and infestation of insects and fungi, which should be considered over the reduction in mechanical properties [21]. Lastly, mechanical properties such as flexural strength, tensile strength, and compressive strength are commonly compared, considering that materials with flexural strength and low density are widely used as beams and girders in structural applications [22]. Considering these three alternatives, it is proposed to compare and assess these materials based on multicriteria decision methods (MCDM). In this sense, the research of Tupėnaitė et al. (2020) [23] used the simple additive weighting method to define the best utilization in high-rise buildings considering the criteria of wood utilization, height of the building, number of floors, building cost, length of implementation, reduction in CO_2_ emissions, and use of energy. The MCDM are driven by a weighted matrix that uses the weight of every criterion to evaluate the candidates. In this sense, the entropy method as a weighting method allows calculation of the degree of difference between criteria and gives a more important value, allowing an objective evaluation [24]. In this way, the research of Reddy et al. (2019) [25] used the entropy method to make a material selection considering sustainable materials for building applications, where 10 sustainable criteria were considered. For the decision maker, this meant a challenge, considering the vagueness and impreciseness addressed by the fuzzy subjective weighting methodology. However, by using the objective criteria, these problems were solved by considering the utilization of the entropy concept. On the other hand, Movaffaghi and Yitmen (2021) [26], used the Complex Proportional Assessment (COPRAS) as the MCDM to study the best timber–concrete composite alternative for buildings. In the same way, Motuzienė et al., (2016) [27] used the COPRAS method to determine the best envelope material for a single-family house, choosing between masonry, log, and timber frames, considering the criteria of reduction in expenses, greenhouse gases, non-renewable primary energy, and ozone layer depletion.

In this way, the purpose of the following research is to characterize the Guayacan timber obtained in the province of Pichincha of Ecuador, manufactured by artisanal production companies, carrying out tests for compressive strength perpendicular to the wood fiber, compressive strength parallel to the wood fiber, flexural strength of wood, tensile strength of wood, moisture content of the wood, and thermal conductivity of the material.

In the same way, a parallel characterization will be performed on oak timber from Loja, El Oro, and Manabi for the same mechanical and thermal tests.

Lastly, the characterized timbers will be compared by multicriteria decision means, using the entropy method as an objective weighting tool and the COPRAS method as the MCDM.

## 2. Materials and Methods

In this section, the testing methodology for thermal conductivity, moisture content, and mechanical characterization will be presented, which includes compressive strength parallel to the grain of the wood, compression perpendicular to the grain of the wood, and resistance flexural and tensile strength. It is important to mention that the tested timbers were obtained from artisanal production, testing the samples in the state that is commonly used. In this sense, even though the density of the wood at a specific moisture content has an important role on the physical properties of the wood, it has not been altered towards the characterization.

### 2.1. Characterization of Ecuadorian Oak and Guayacan Timber

Different samples from different locations are studied considering that small and clear specimens of the same species can vary their mechanical properties, such as in the case of the modulus of elasticity, which could change in a range of approximately 7–20% [28]. In this sense, thermal and mechanical tests were carried out with hard oak wood specimens and Guayacan timber. The oak hardwood was obtained from three different provinces of Ecuador in the Coastal region: Manabí, El Oro, and Loja. In the case of the Guayacan timber, it was obtained from artisanal producers in the province of Pichincha in the highland region, as shown in Figure 1.

### 2.2. Thermal Tests

For the thermal tests, 12 specimens were used: 3 specimens of oak from El Oro, 3 specimens from Manabí, 3 specimens from Loja, and 3 specimens of Guayacan from Pichincha. The tests for the determination of thermal conductivity are based on the ISO 8302:1991 standard [29] and ASTM-C177 [30] by the sheltered hot plate method, where the samples must have a homogeneous composition with an area of 150 × 150 mm² and thicknesses from 10 mm to 200 mm. The tests are carried out at temperatures between −10 °C and 50 °C. This procedure allows measuring conductivities ranging from 0.002 to 2.5 W × m^−1^ × K^−1^. The specimens must previously be kept in a room with humidity and a controlled temperature of 23 °C for 24 h. After the necessary conditioning time has elapsed, measurements are taken with precision scales of ±0.05 g. It is verified with precision instruments that the sample complies with the ISO 8302 standard; then, 5 measurements of the specimen are taken to have an average or arithmetic mean. Then, the samples are covered with polypropylene plastic and placed on top of the ultrasonic gel sample, avoiding vacuum spaces.

Furthermore, to measure the thermal conductivity of the test samples, the hot plate thermal machine, Lambda-Messtechnik λ-Meter EP500e Version C from the thermal testing and energy efficiency laboratory of the Escuela Politécnica del Litoral (ESPOL), was used and displayed in Figure 2, along with Equation (1):(1)$$k=\frac{q\times L}{∆T\times A}$$
where q (W) is the heat flow, k (W × m^−1^ × K^−1^) is the thermal conductivity (m^2^), A is the area of the section, ΔT is the difference of temperatures through the sample, and L (m) is the thickness of the sample.

### 2.3. Mechanical Test

For the mechanical tests, 80 specimens were needed, which were divided for the different tests of parallel compression, perpendicular compression, flexion, and tensile, in such a way that 5 samples from every location were used, taking into consideration the limitations for using the measurement equipment. Every sample was tested with the standard ASTM D143-14 [31] under the methodologies for parallel compression, perpendicular compression, flexion, and tension parallel. The tests were carried out with the WAW- 6OOC Computer Control Electra-hydraulic Servo-controlled Universal Testing Machine from the Particular International University (SEK), as shown in Figure 3.

#### 2.3.1. Parallel Compression

The dimensions of the specimens for this test are 20 cm long with a rectangular section of 5 × 5 cm, considering that a prism must be perpendicular on its faces. A tolerance of ±1 cm is considered, even if that is not part of the recommendations of the standard. The speed of application of load is 0.6 mm/min until the break occurs, and the failure stress to compression is obtained with Equation (2) [31]:(2)$$\sigma_{c}=\frac{F_{c}}{A}$$
where σ_c_ is the failure compressive stress parallel (MPa), F_c_ is the failure load (N) in compression, and A is the cross-sectional area of the sample.

The calculation of the modulus of elasticity uses Hooke’s law considering σ_c_ as the parallel compressive stress (MPa), E is the modulus of elasticity or Young’s modulus, (MPa) and ε is the unitary deformation, according to Equation (3), while Figure 4 shows how the test was conducted.
(3)$$E=\frac{\sigma_{max1}}{\varepsilon }$$

#### 2.3.2. Perpendicular Compression 

The test specimen is a prism with a rectangular section of 5 × 5 cm and a length of 15 cm that must be cut with these exact measurements. A metallic plate with dimensions of 5.11 × 11.6 cm with a thickness of approximately 0.5 is used to distribute the load to the center of the sample, in such a way that the failure occurs by crushing. The continuous load speed applied is 0.305 mm/min until it reaches failure or a maximum penetration of 2.5 mm. The resistance to the limit of proportionality–RLP (MPa) is calculated with Equation (4) [31]:(4)$$RLP=\frac{F_{lim}}{A_{c}}$$
where F_lim_ is the load at the proportional limit (N) and A_c_ is the contact area by the steel plate on the surface of the sample. On the other hand, the maximum unit resistance is calculated with Equation (5):(5)$$\sigma_{c}=\frac{F_{p}}{A_{p}}$$
where σ_c_ is the maximum unitary resistance to compression (MPa), F_P_ is the necessary load up to penetration of 2.5 mm (N) and A_p_ is the contact area by the plate steel on the surface of the sample. The sample tested for perpendicular compression is displayed in Figure 5.

#### 2.3.3. Bending Test

The test sample is a rectangular section prism 5 × 5 cm and 76 cm long, with ±3% tolerance. Five measurements are taken from the sample on the central part to obtain an average. The distance between supports must be 71 cm and a support must allow rotation in the axis of the test sample. The speed for this test is 2.5 mm/min until the test piece breaks with a limit load F_lim_ on the proportional limit, while the unit stress in the proportionality limit–σ_UL_ (MPa) is calculated with the following Equation (6). The test is displayed in Figure 6.
(6)$$\sigma_{UL}=\frac{3F_{lim}\times L}{2b\times h^{2}}$$
where L is the length between supports (mm), b is the base of the specimen (mm), h is the height of the test sample (mm) and F_m_ is the maximum load. The maximum unit stress to flexion–σ_b_ (MPa) is calculated with Equation (7):(7)$$\sigma_{b}=\frac{3F_{m}\times L}{2b\times h^{2}}$$

The modulus of elasticity on the unit proportional limit–E_lim_ (MPa) is calculated with Equation (8) and the modulus of elasticity–E (MPa) with Equation (9), where d_1_ (mm) is the displacement of the test sample on the proportionality limit, σ_b_ (MPa) is the stress on the proportionality limit, and ε is the percentual unitary deformation.
(8)$$E_{lim}=\frac{F_{lim}\times L^{3}}{2d_{1}\times {b\times h}^{3}}$$
(9)$$E=\frac{\sigma_{b}}{\varepsilon }$$

#### 2.3.4. Tension Test

The test is performed by the application of an axial tensile force until failure. The dimensions of the specimens had a length of 46 cm, and the central section was 0.98 × 0.45 cm for 6.35 cm. The end supports had an extension of 10.1 cm and will have a square section measuring 2.54 × 2.54 cm. The speed for this test is 1 mm/min continuously until failure of the specimen. The maximum unit stress to tension–σ_t_ (MPa) is calculated with the area of the minimum section A_min_ (m^2^) in the sample following Equation (10) and the unitary stress on the limit of proportionality–σ_lim_ is calculated using the tensile load F_t_ with Equation (11) and Figure 7 shows a sample being tested:(10)$$\sigma_{t}=\frac{F_{max}}{A_{min}}$$
(11)$$\sigma_{lim}=\frac{F_{t}}{A_{min}}$$

### 2.4. Moisture Content (MC) Tests

Water is present in the wood in three forms: hygroscopic or fixed, free water that fills the cell cavities within the wood, and bound water that is held by cellulose. In this sense, the moisture content test was carried out under the standard ASTM D4442 [32] considering that to determine the content of humidity, a relationship is established between the mass of free water and the mass of hygroscopic water contained in a sample, concerning the mass of the anhydrous sample (oven dry weight) [33]. The preparation of the specimens is later for each mechanical test; hence the number of test pieces was 80. The specimens must be cut and acquired from the nearby area to failure and stored under conditions that do not alter the content of humidity. The specimens were labeled for recognition and subsequently weighed before putting them in the oven. Drying time is 24 h at a temperature of 103 °C ± 2 °C. After this time, proceed to remove the specimens from the oven for cooling to room temperature of 20 °C ± 2 °C, before being weighed for the last time. The determination of the moisture content of wood is achieved by following Equation (12):(12)$$MC=\frac{w_{w}\times w_{d}}{w_{d}}\times 100\%$$
where MC is the moisture content (%), w_w_ is the weight of the wood in its initial wet state (g), and w_d_ is the weight of wood in a dry state (g). Lastly, Figure 8 shows all the tested samples submitted to the MC analysis.

### 2.5. Multicriteria Decision Methods

Previously, we had studied the utilization of alternative materials for building applications and the selection of the best material by MCDM, where it was found that the bamboo of Ecuador, also known as Guadua, presented the best properties regarding resistance as beams for social-interest housing [34]. In this sense, the same material was compared to Guayacan and the oak from different zones of origin in Ecuador. However, in this case, the thermal conductivity is considered as a factor for temperature insulation. In this sense, the following Table 1 displays the properties of the 5 materials to be assessed by MCDM, which is considered a matrix of i rows and j columns. Information was selected by CES-Edupack; Granta Design Limited [35].

In this way, considering that the materials present different capabilities for the different samples, the present selection took the worst scenarios, showing that the material with the lowest thermal conductivity is Guayacan, making it the best for insulating the environmental temperature. The highest value is the worst, considering that the material will conduct external heat into the building; hence, the worst insulating material is the oak of Manabi. The flexural modulus has an important advantage for the oak of Loja and the worst performance for bamboo. The compressive strength is the best in the case of bamboo and the worst for Guayacan. In terms of bending strength, the oak from El Oro shows the best properties in contrast to the bamboo which is the worst, while in terms of the tensile strength, the oak of El Oro again shows the best performance and Guayacan, importantly, falls behind. In this sense, there is no optimal material, and the performance changes depending on the material’s characteristics, which makes it suitable to use the MCDM to assess the best material.

#### Entropy-Weighted COPRAS Method

The research of Shi and Sun (2023) [36] considered that the entropy method delivers an objective evaluation of the weights of each criterion reflecting the importance of the information delivered. In this sense, the entropy method follows the subsequent steps.

The decision matrix is generated based on the values of every material characterized, via Equation (13), where the *X_ij_* values represent every position of Table 1; for example, X_11_ represents the thermal conductivity value of the material M1.
(13)$$P_{ij}=\frac{x_{ij}}{\sum_{j=1}^{n}x_{ij}}$$

These criteria are normalized and then calculated in entropy Equation (14).


(14)
$$E_{i}=\frac{\sum_{j=1}^{n}P_{ij}\times \ln P_{ij}}{\ln n}$$


2.The last step weighs each criterion with Equation (15) [37].


(15)
$$w_{i}=\frac{1-E_{i}}{\sum_{i=1}^{m}\left(1-E_{i}\right)}$$


On the other hand, the ranking of the COPRAS method is developed based on the decision or objective matrix (DOM) and evaluates the alternatives considering their relative importance [38]. The present method was guided by the following steps:

The normalization matrix is reached with the original DOM using the summation for normalization, as shown in Equation (16).
(16)$$r_{ij}=\frac{X_{ij}}{\sum_{i=1}^{m}X_{ij}}$$

The weighted matrix is found by multiplying each objective value by the weighted value found in the entropy method, as shown in Equation (17).


(17)
$$y_{ij}=r_{ij\;\;}\times W_{j}$$


2.The weighted normalized scores are calculated considering the maximization of the objectives with Equation (18). This means that for all the criteria a higher score is better, except for the thermal conductivity where a lower score is better. Equation (19) refers to minimizations.


(18)
$$S_{+i}=\sum_{j=1}^{n}y_{+ij}$$



(19)
$$S_{-i}=\sum_{j=1}^{n}y_{-ij}$$


3.The calculation of comparative significance Qi determinates the relative importance of each alternative, as shown in Equation (20).


(20)
$$Q_{i}=S_{+i}+\frac{\Sigma_{i=1}^{m}S_{-i}}{S_{-i}\sum_{i=1}^{m}{\left(S_{-i}\right)}^{-1}\;}$$


4.The level of choice utility is found at last (in Equation 21), which makes it possible to rank the alternatives [38,39].


(21)
$$U_{i}=(\frac{Q_{i}}{Q_{max}})\times 100\%$$


## 3. Results and Discussion

### 3.1. Thermal Characterization of Guayacan and Ecuadorian Oak

The thermal conductivity results of the Ecuadorian oak (Figure 9) showed that in eight of the nine specimens, the thermal conductivity has a value between the range of 0.16 and 0.19 W × m^−1^ × K^−1^. This range is defined for the test conditions according to the ASTM-C177 standard. In this way, test 2 does not comply with the standard of thermal conductivity since it showed a conductivity of 0.142 W × m^−1^ × K^−1^ established in the values of the ASTM-C177 standard [30]. In the same way, for Guayacan timber, the thermal conductivity average was found to be 0.105 W × m^−1^ × K^−1^. Figure 9 compares all samples while the numerical results are displayed in the appendix. The thermal conductivity is greater for the oak rather than the Guayacan, while in the differentiation of the different zones from its origin, there is a slight difference that gives an advantage to the oak from Manabi.

### 3.2. Mechanical Tests of Ecuadorian Oaks and Guayacan Timber

The results of the mechanical test have been unified in Figure 10 where a comparison of the Guayacan timber vs. the Ecuadorian oaks from three different zones of the country can be seen, showing the strength against perpendicular compression, bending, and tension following the left x-axis and the modulus of elasticity from parallel compression in the right x-axis. It is important to mention that the modulus of elasticity has been calculated for both parallel compression and bending; however, the bending test was performed following the three bending configurations, which is not purely axial, and compared to the four points test it underestimated about the 19% of the modulus of elasticity [40]. For this reason, the modulus of elasticity from the parallel compression test is displayed. In the same way, the data obtained are displayed in Appendix A.

The five test samples of guayacan wood were subjected to compression with loads exceeding 45 kN. The samples G2PC and G5PC have a lower failure stress with a difference of 1.2835 MPa based on the standard deviation. On the other hand, for the five test specimens subjected to perpendicular compression by crushing with loads from 6.81 kN to 7.35 kN, an average of 7.19 (MPa) is recorded for the maximum compression stress resistance by crushing, with a standard deviation of 0.32 MPa which indicates that the variation is low, showing that all the test samples have the correct dimensions based on the standard. The flexural resistance test shows that the sample G2F has the greater resistance based on the applied load of 8.24 kN, while in the case of the tensile strength test, there is a variation of the load between the specimens due to dimensions that are not exact for this type of tests. However, the results show an average for maximum stress of 79.03 MPa, and the standard deviation indicates a variation of 14.62 MPa due to the variation of loads, possibly resulting from the moisture content that reduces the resistance of the samples.

On the other hand, the mechanical properties of the same oak species obtained from different parts of Ecuador were characterized, showing that the elastic modulus values are higher in the case of the oak that comes from the province of Loja and in the case of El Oro province, 40% of the test pieces did not meet the results according to the ASTM D143 standard [31]. Considering the results of the average maximum resistance stress to perpendicular compression σ_c_, the tests comply with the ASTM D143-14 standard [31]. In the 15 specimens tested from the three regions, the maximum strength (MPa) varies between 19.08 and 20.69 MPa, with better performance for the oaks from Manabí, but without an important difference. On the other hand, the bending tests of the oak hardwood show a result ranging from 77.11 to 89.80 MPa, complying with the ASTM D143-14 standard [31]. The results obtained show how the oak from El Oro province is higher than those of other provinces, but considering the standard deviation, the oak from the coast region (El Oro and Manabi) could behave in the same way, while there is a slight difference for Loja. The results of the tensile tests showed that samples O3T-Manabi, O3T-Loja, and O5T-Loja had values below the reference values of the ASTM D143-14 standard, while all the samples coming from the province of El Oro complied with the standard [31]. In the same way, the oak from El Oro showed the best performance with an average tensile strength of 189.41 MPa but again considering the standard deviation, it is no different from the Manabi samples.

Comparatively, traditional Chinese structural timber has served as the main material for the ancient buildings of the region, making them one of the most famous building systems in the world for its configuration in joints. This traditional building has been characterized and was found to have a compressive strength of 30 MPa, a bending strength of 50 MPa, and a tensile strength of 91.4 MPa [41] which is comparable with the Guayacan timber, except for the compressive strength where it shows a deficiency. On the other hand, different research has studied alternative materials for building, intending to face environmental challenges. Within these materials, the development of compressed wood (CW) can be found, which is manufactured by thermos–mechanical compression, allows a substantial increase in the mechanical properties, making them competitive against other structural materials [42]. In this sense, the research of Haller and Wehsener (2004) [43] showed a bending and tensile strength of 169 Mpa and 185 MPa, respectively, which is comparable in the case of the tensile strength of the Ecuadorian oak, but not for the bending strength.

### 3.3. Moisture Content of Ecuadorian Oaks and Guayacan Timber

The weight of the specimens in the initial wet state and the anhydrous state (final weight), extracted from the samples of parallel compression, along with the average moisture content, are displayed in the appendix while a comparative graphic compares its behavior in Figure 11. It is observed that specimen G5MC has a significant reduction in moisture content compared to the other samples, while specimen G3MC has a higher moisture content for Guayacan, which is the material with the lowest moisture content compared to the oaks. On the other hand, the moisture content has a higher moisture content, but the sample O5MC-ElOro is shown to be higher. On average, the oak from Manabi shows a lower MC% among the oaks, a fact that may explain the best thermal conductivity. In the case of moisture content, none of the specimens of oak hardwood exceed 30%, as established in the ASTM D4442 standard [32].

In normal environmental conditions, the moisture content can be a dangerous factor for building materials, considering that excessive moisture can lead to deteriorated habitation quality, and a reduction in the resistance to high temperatures. There are more possibilities to increase the mechanical stress of the material, salt transportation, and decay [44]. These considerations make Guayacan a material to be considered, regarding its low moisture content in comparison to the oak of all other regions. Furthermore, this anormal moisture variation shows that the artisanal method is not the best and should be improved.

### 3.4. Comparison by Entropy-COPRAS MCDM 

The comparison regarding MCDM began with the weighting of all criteria. As an objective method, entropy shows that the thermal conductivity is the most important characteristic, with a 0.99 entropy and a weight of 0.212, which is congruent with our study considering that the efficiency of thermal insulation depends on the thermal conductivity since this property allows the material or the combination of materials to slow down the heat flow from conduction, convection, or radiation origins [44]. On the other hand, regarding the mechanical properties, the bending strength is placed second but slightly over the tensile strength and with a more important weight compared to the modulus of elasticity. The comparison shows that the difference in thermal conductivity between candidates was considered as a more important factor [25,36], ensuring that the weight of the criteria has the expected objective component. The results of the entropy weighting method are presented in Table 2, obtained by the utilization of Equations (12)–(14).

Since the criteria weights have been delivered by the objective entropy method which gives a weight by mathematical means without the subjective intervention of the selector, the MCDM COPRAS allows to rank the materials, a step towards finding the best material. In this sense, even though thermal conductivity is the most important criterion and Guayacan presents the best property for insulation systems, its deficiency in terms of mechanical properties puts it at last with a significant difference, with a utility of 61%. On the other hand, even when compared to other cement-reinforced materials, bamboo proved to be the best [34]; compared to the characterized materials, it showed a utility of 86% which is not far from the oak of Manabi, which is the next best, with a utility of 88%. However, the oak from Loja is considered the best and shows a significant advantage over the oak from El Oro which is the second best. These results can be seen in Table 3.

Even when the oak from Loja is not the best in every criterion, it stands as the best for its modulus of elasticity, since in all the criteria there is not much differentiation but in the modulus of elasticity the oak of Loja shows an outstanding superiority. Objectively, all the materials have a building application that can be comparable to traditional ancient buildings [41] but oak can be considered the best among them. With an intrinsic characteristic of modulus of elasticity, it would be preferable to use the oak from Loja.

## 4. Conclusions

The research on climate change needs to explore all the alternatives to prevent greater damage to the planet. In this sense, the present research has characterized two kinds of woods: Guayacan and oak from Ecuador, to give an alternative to the building sector.

In this way, the results show that Guayacan has lower mechanical resistance and thermal conductivity. The material itself has these disadvantages that could be related to a significantly lower moisture content.

On the other hand, the oaks have shown good properties that can change depending on the origin zone of the wood. It was determined that the oak from Manabi has a slight advantage in thermal conductivity, which at the same time could be related to a lower moisture content. However, regarding the mechanical properties, there is not much difference for the compression, bending, and tensile strength, nevertheless, for the modulus of elasticity, the oak from Loja stands out, making this a factor to be considered in the selection of a wood for building applications.

Considering that different materials show different properties, in the case of the same species but from other regions there are similitudes. This makes it important to study new species with the appropriate standard.

Moreover, timbers can be used for different applications that require different properties, making the entropy method a good technique for assessing the weight of the criteria objectively and for general purposes. In this sense, the multicriteria decision methods driven by weighted objective techniques show that an assessment that can lead to better selection by mathematical means can be performed. This is the case when using an entropy-COPRAS method that has not been thoroughly explored for building material selection. In this sense, this research allows us to analyze a different approach for material selection for buildings.

On the other hand, the materials were studied considering their artisanal production. It is recommended to continue the research with a more technical treatment of the material, taking into account the variation of moisture presented.

## Figures and Tables

**Figure 1 biomimetics-09-00443-f001:**
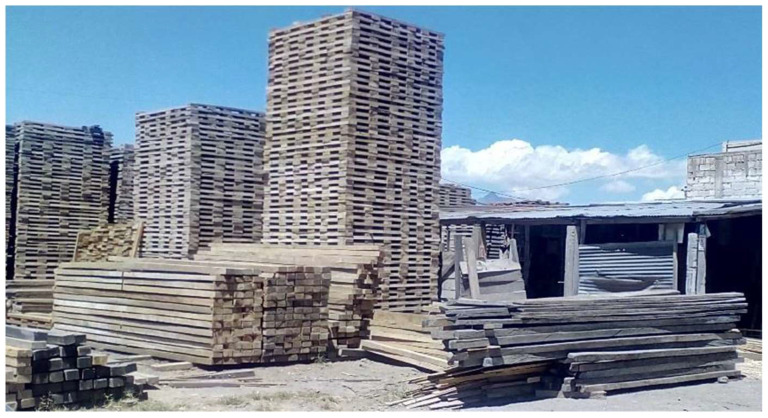
Artisanal production of timber.

**Figure 2 biomimetics-09-00443-f002:**
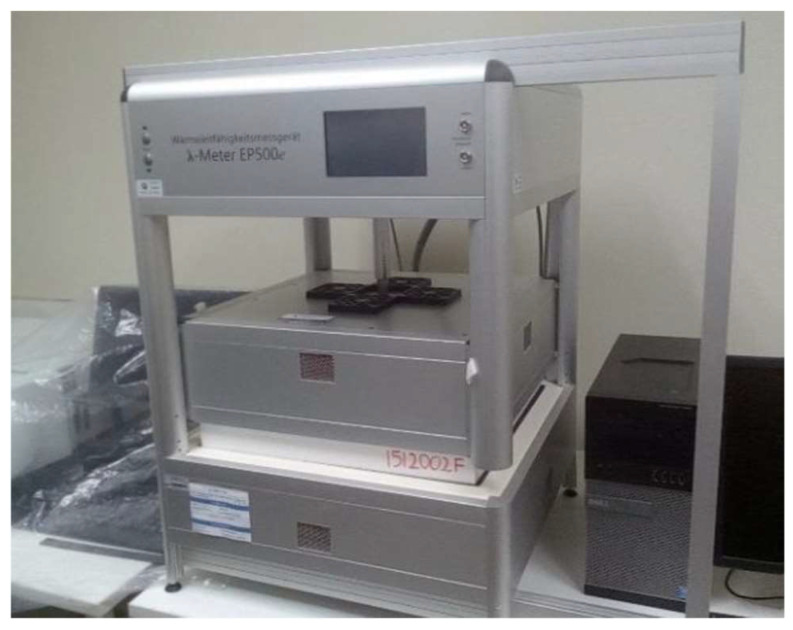
Thermal conductivity test machine.

**Figure 3 biomimetics-09-00443-f003:**
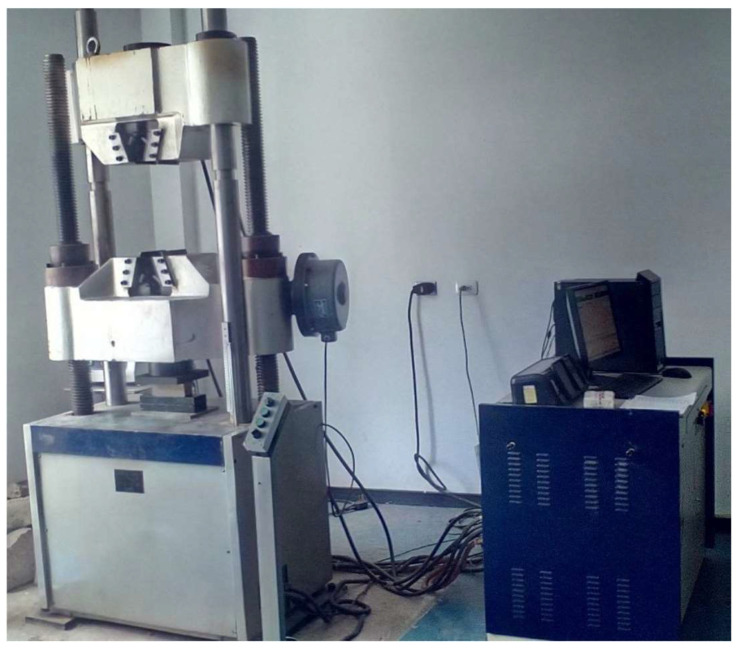
Universal testing machine.

**Figure 4 biomimetics-09-00443-f004:**
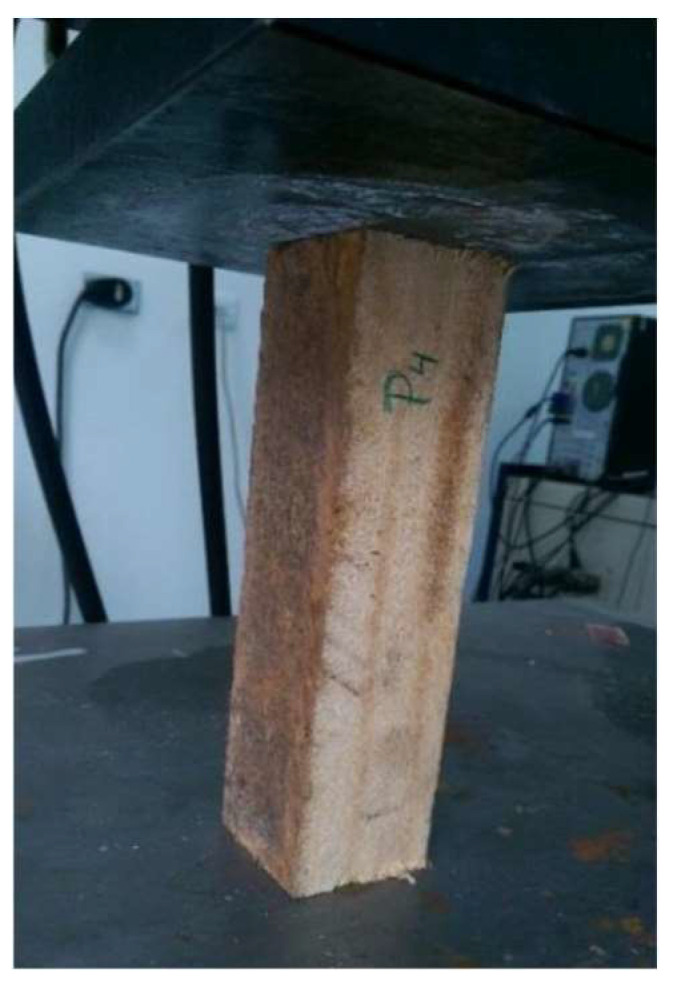
Parallel compression test.

**Figure 5 biomimetics-09-00443-f005:**
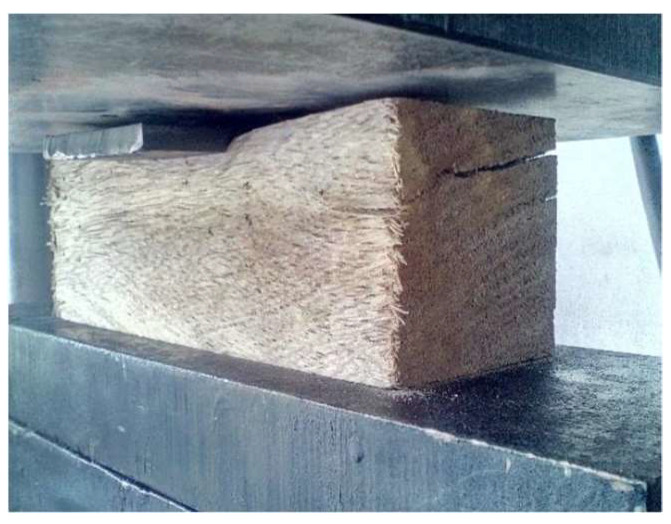
Perpendicular compression test.

**Figure 6 biomimetics-09-00443-f006:**
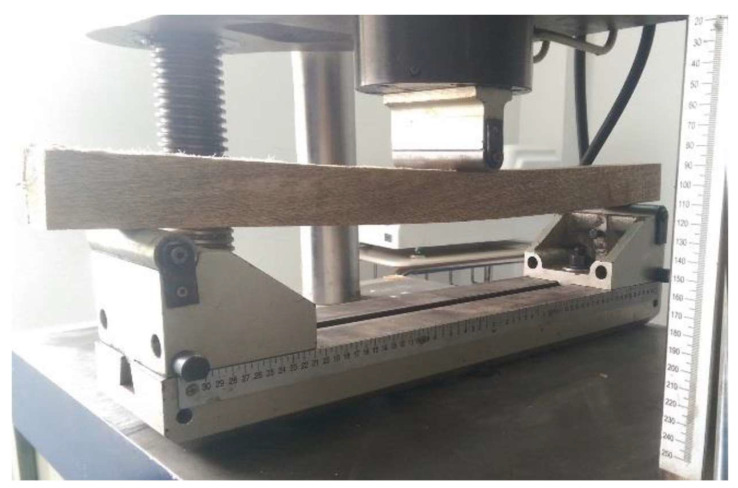
Bending test.

**Figure 7 biomimetics-09-00443-f007:**
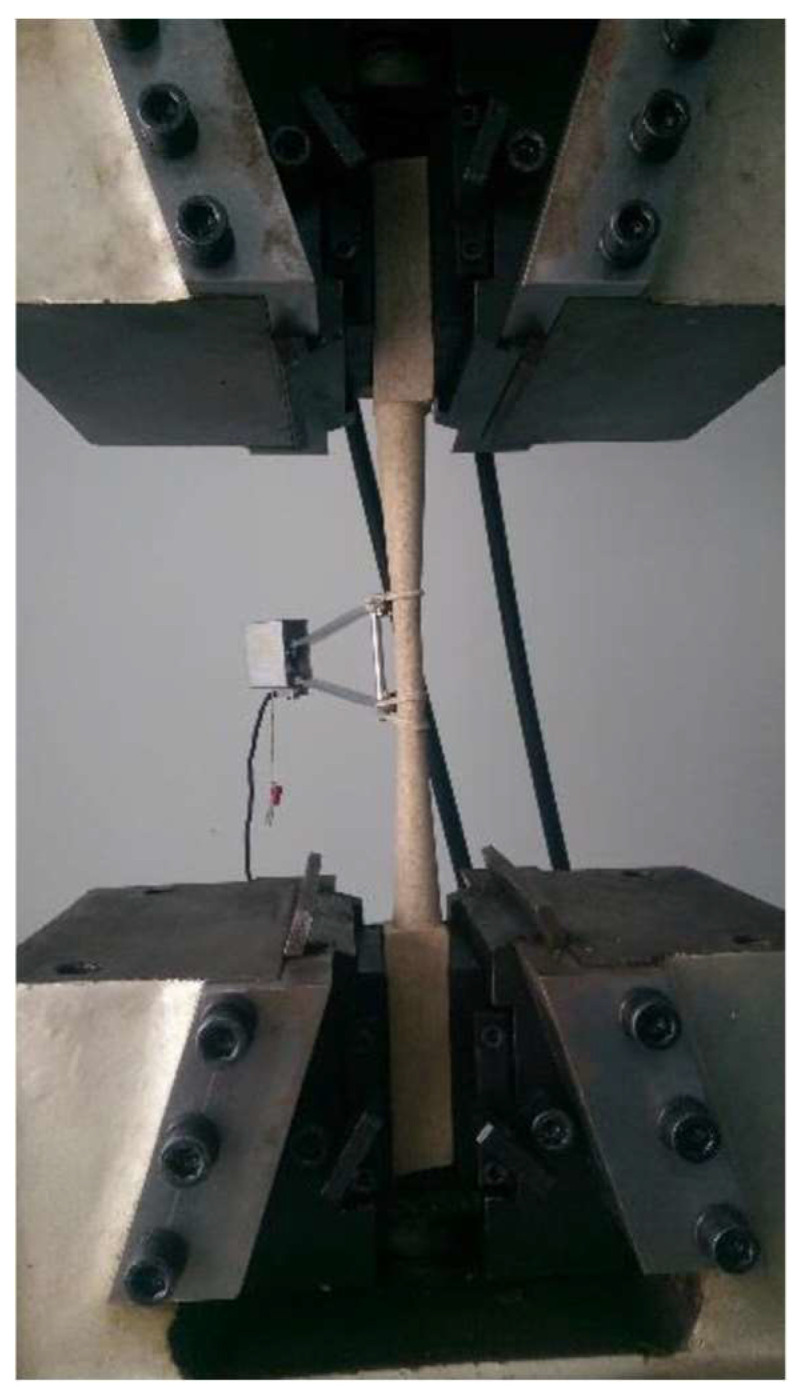
Tension tests.

**Figure 8 biomimetics-09-00443-f008:**
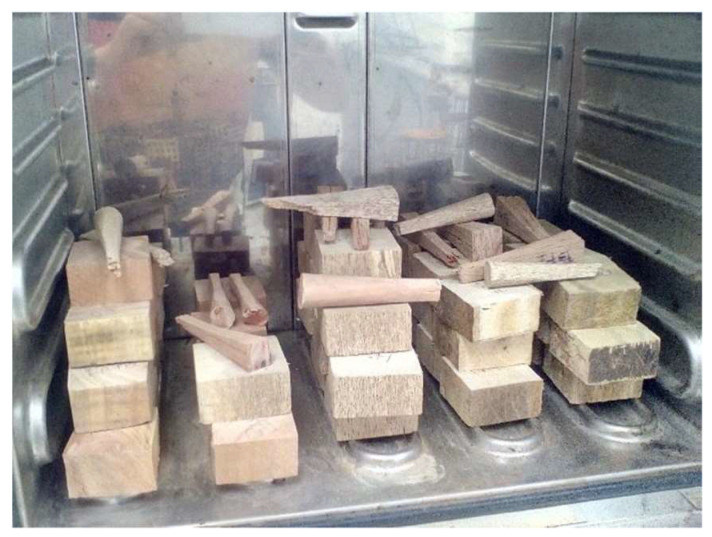
MC tested samples.

**Figure 9 biomimetics-09-00443-f009:**
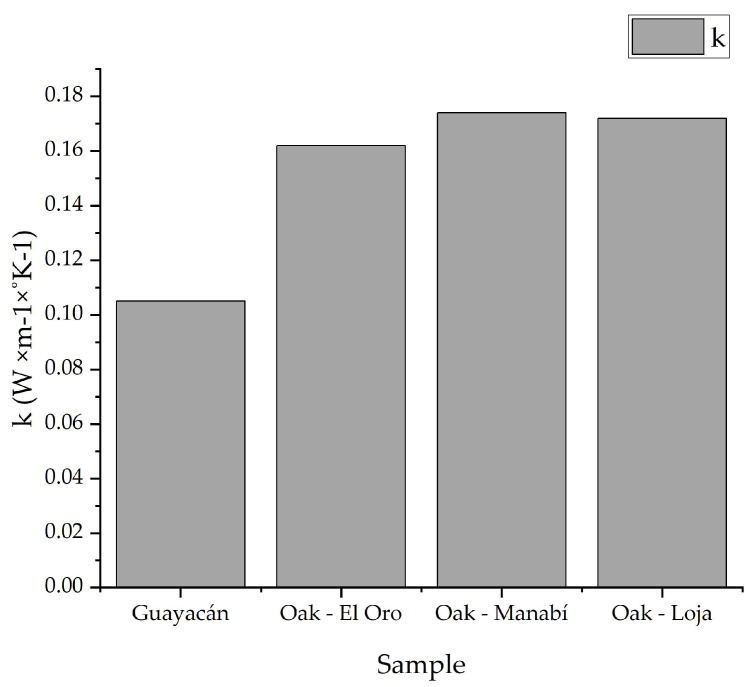
Thermal conductivity comparison.

**Figure 10 biomimetics-09-00443-f010:**
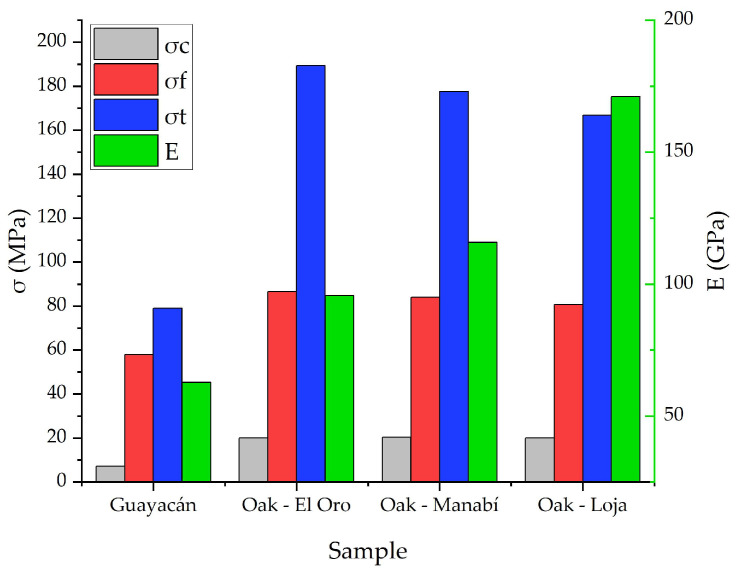
Mechanical properties.

**Figure 11 biomimetics-09-00443-f011:**
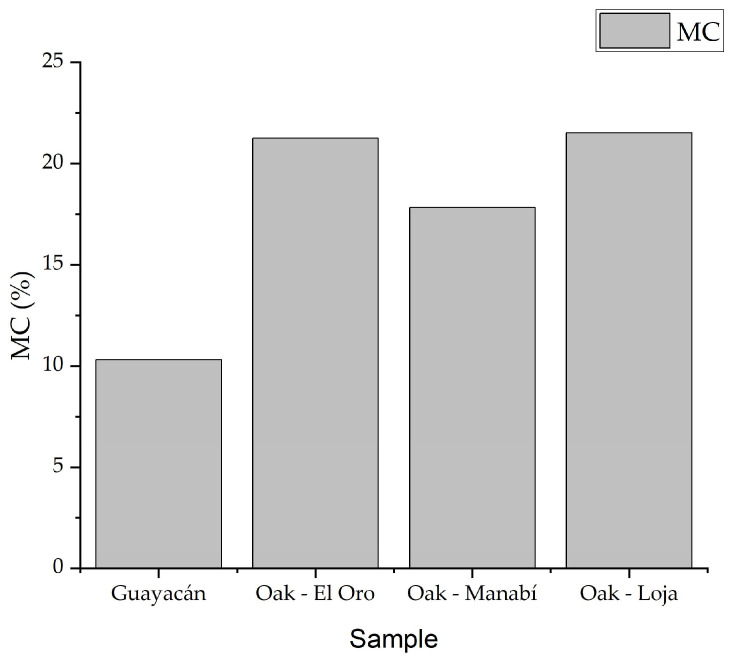
Moisture content comparison.

**Table 1 biomimetics-09-00443-t001:** MCDM materials properties.

Material	Index	Thermal Conductivity	Flexural Modulus	Compressive Strength	Bending Strength	Tensile Strength
		(W × m^−1^ × K^−1^)	(GPa)	(MPa)	(MPa)	(MPa)
Bamboo *	M1	0.18	17	60	35.9	160
Guayacan	M2	0.105	36.2	6.81	54.94	64.81
Oak El Oro	M3	0.162	72.4	19.23	82.56	173.54
Oak Manabi	M4	0.176	91.36	20.12	77.11	158.13
Oak Loja	M5	0.172	151.36	19.08	77.62	148.75

* Properties of the material extracted from software [35].

**Table 2 biomimetics-09-00443-t002:** Entropy method result.

Criteria	Entropy	Weight	Rank
Thermal conductivity	0.99	0.212	1
Modulus of elasticity	0.87	0.187	4
Compressive strength	0.86	0.184	5
Bending strength	0.98	0.209	2
Tensile strength	0.97	0.208	3

**Table 3 biomimetics-09-00443-t003:** COPRAS method.

Material	Qi	Ui	Rank
Bamboo	0.203	86%	4
Guayacan	0.144	61%	5
Oak El Oro	0.209	89%	2
Oak Manabi	0.208	88%	3
Oak Loja	0.236	100%	1

## Data Availability

Will be provided on demand.

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
