# Peer review of "Ecuadorian Woods: Building Material Selection Using an Entropy-COPRAS Comparative Analysis Based on the Characterization of Ecuadorian Oak and Guayacan Timber"

_biomimetics, 2024, doi:10.3390/biomimetics9070443_

Round 1

Reviewer 1 Report

Comments and Suggestions for Authors

The authors investigate the properties of four Ecuadorian wood species using an Entropy-COPRAS comparison. In fact, the manuscript contains two investigations: physical testing of wood material of four different species/origin and a comparative evaluation. The manuscript contains valuable parts, while some parts are not sufficiently elaborated or some necessary explanations missing. The authors should address several major and minor issues to improve their work as detailed below.

Major concerns:

1. The material tests are not sufficiently documented and elaborated. It would be advisable to illustrate the experiments with photographs. Though numerical results of the tests are given in tables, stress-strain / load-deflection diagrams also have to be added, one for each test type. These diagrams would enable the comparison of different species and also show the characteristic behaviour.

2. Compression tests parallel to grain.
It is not clear how the authors interpret 'break' and 'ultimate stress'. Timber specimens can exhibit various failure modes, and full width kinking is one of the typical and important modes. The stress-strain diagram has a maximum stress value but it does not mean breaking in the strict sense because the onset of kinking is followed by a steady-state deformation at lower stress level. For more on the characterisation of the stress-strain diagrams in compression, the authors may consult e.g. Saad (2024) Int J Arch Heritage, 1-11, or André (2014) Constr Build Mat, 50, 130-139. The authors should clarify whether 'ultimate' refers to the maximum or the steady-state. Presenting the related stress-strain diagrams as mentioned in Comment 1 would also serve well to clarify the issue.
Also, a variation of +-1cm is mentioned, which does not seem to be in accordance with the standard recommendations.
Another important issue is the calculation of the modulus of elasticity E. It is supposed to be applied to the initial section until the proportionality limit was reached. It is not specified how it was obtained.

3. Compression test perpendicular to grain.
The authors mention that the specimen must be homogeneous. It is somehow wierd because wood is naturally inhomogeneous. Do the authors mean that it should not contain knots or defects? Please clarify.
Proportionality limit is also mentioned here. The same comment applies as above.

4. Bending.
The bending tests shown here seem to be 3-point bending. It is acceptable that the authors follow the standards but it is worth considering that in 3-point bending the stress state is not a purely axial one but a complex axial+shearing stress state develops and the validity of the bending stress results is somewhat compromised. (The reviewer conducts 4-point bending for own test for this reason.)
Another issue is that equations 6 and 7 seem to be identical. Please correct.
Also, the modulus of elasticity is computed here in addition to the parallel compression section. Which one is shown in the tables, from compression tests or from bending?

5. The physical testing and evaluation can be an important contribution if it involves new species not investigated previously, therefore it would be important to correct and amend this part of the manuscript as mentioned above.

6. Entropy weighted COPRAS method.
I have some concerns regarding this method. The discription of the method is lacking. At the start, in equations 13 and 16 terms xij are not defined. The major problem is that it is completely missing what input data the user should define and on what basis? Is it the xij values and how are they determined? The method is supposed (as it seems) to rank the alternatives on some basis but nothing is shown about it. The authors write (L351) that "Since the criteria weights has been delivered by the objective entropy method, ..." Does the term "objective" refer to that it gives objective results without user-specified parameters, or does it refer to the objective of a structural design, or something else? It is not shown how the entropy values in Table 2 are obtained. To me the physical criteria listed in Table 2 seem user-specific, i.e. they depend on the objectives of the design. For example, in bridge design bending strength and flexural modulus are important while thermal conductivity is irrelevant, or in apartment buildings the thermal conductivity has larger importance while mechanical properties also matter, etc. As the ranking is one of the authors' conclusions, these issues need to be clarified.
Also, as it seems, the method is only applied in this work without any original contribution, or if any part is the auhors' development, it is not clear to me.

7. Conclusions.
The conclusions on ranking heavily depend on the evaluation of the method as described above. As it is not shown how the results are obtained, it is also not known how sensitive the results are to the choice of the input values (whatever they are). Moreover, the authors write that the results for Guayacan can be the consequence of its low moisture content. In fact, all investigated species have quite different moisture content, which makes the comparison less reliable. The authors may consider how the moisture content would affect the mechanical properties and thus the ranking as well. Therefore, these conclusions can be considered only as a starting point for further investigations, but definitely not exhaustive as they are. However, numerical results from the experiments have value and can be important.

Minor issues

1. The use of technical terms is often wrong, inconsistent or inaccurate. The word "module" appears several time (L183, L272, etc), supposedly meaning "modulus" but "module" has a completely different meaning in English. I recommend to use the term "modulus of elasticity" throughout the text. The authors write the term "test tube" a few times (L156,L175) but I have found no reference to it in the standard; if it is a reference to the test specimens, it is wierd as they are rectangular. The term "deformation" is used (L184) for d1, which is clearly not a deformation. I recommend the term "deflection" or "displacement". The term "summatory" (L244) is an adjective, so use the noun "summation". The term "charge" (L283) seems odd, mistyped maybe?

2. The figures display values for independent entities (wood species), so connecting the dots with solid lines makes no sense. It is advised to use bar plots or similar for such data.

3. Mistyped value in Table A4, column E, last row. Std should be 7.621.

4. Clarity of some sentences is problematic. Sentence in L74-81 is too long and difficult to understand. The same for L90-94. Sentence in L287-288 has no verb in it. Etc.

5. Referencing is not uniform. Use [number] consistently. Remove Name(year) in L90, L94-95, L97, and keep [number].

6. Provide the Latin names for the wood species investigated in this work. I can see the Guayacan in L58 but missing the Oak.

7. Use capital initials (F,T) for figures and tables in several places in the text, e.g. ...shown in Figure X.

8. Label G2P5 (L275) seems to be mistyped.

9. The use of English has a lot of minor errors, some typos but mostly grammatical errors. It includes inconsistent subject-verb matching, wrong word order, missing terms, etc. A thorough check and corrections are necessary.

Overall assessment: I find the physical testing part reliable and important and the authors are encouraged to extend their experiments with larger samples in the future. The comparative analysis is not sufficiently established and presented, and I find the related conclusions only indicative in the lack of much larger corpus of test data. The authors should address the abovementioned issues and revise the manuscript to warrant publication.

Comments on the Quality of English Language

Several minor issues as mentioned in the review. A thorough check and corrections required.

Author Response

Dear reviewer,

Thank you for the time and work spent in the revision of our paper, it is very important for us to be considered for publication.

Regarding your observations.

  1. The material tests are not sufficiently documented and elaborated. It would be advisable to illustrate the experiments with photographs. Though numerical results of the tests are given in tables, stress-strain / load-deflection diagrams also have to be added, one for each test type. These diagrams would enable the comparison of different species and also show the characteristic behaviour.
    1. Figures 1 to 8 has been added to have a better documentation. The behaviour of the tested materials has been changed to a bar diagram in figures 10 to 11 for a better comparison
  2. Compression tests parallel to grain.
    It is not clear how the authors interpret 'break' and 'ultimate stress'. Timber specimens can exhibit various failure modes, and full width kinking is one of the typical and important modes. The stress-strain diagram has a maximum stress value but it does not mean breaking in the strict sense because the onset of kinking is followed by a steady-state deformation at lower stress level. For more on the characterisation of the stress-strain diagrams in compression, the authors may consult e.g. Saad (2024) Int J Arch Heritage, 1-11, or André (2014) Constr Build Mat, 50, 130-139. The authors should clarify whether 'ultimate' refers to the maximum or the steady-state. Presenting the related stress-strain diagrams as mentioned in Comment 1 would also serve well to clarify the issue.
    1. According to the referred literature, it has been made a distinction on the maximum stress supported and changed the term “utimate” for failure on lines 193; 195; 371
  3. Also, a variation of +-1cm is mentioned, which does not seem to be in accordance with the standard recommendations.
    1. It has been stablished that a tolerance of +-1cm is considered even if that is not part of the recommendations of the standard on line 192
  4. Another important issue is the calculation of the modulus of elasticity E. It is supposed to be applied to the initial section until the proportionality limit was reached. It is not specified how it was obtained.
    1. the modulus of elasticity for the compression parallel to grain is calculated with the failure stress to compression calculated in eq 2. The equation and determination of the calculation has been corrected on lines 192 to 200
  5. Compression test perpendicular to grain.
    The authors mention that the specimen must be homogeneous. It is somehow wierd because wood is naturally inhomogeneous. Do the authors mean that it should not contain knots or defects? Please clarify.
    Proportionality limit is also mentioned here. The same comment applies as above.
    1. Referring to homogeneous we were trying to say that the prism must have the exact measures, this has been changed in line 205

  6. The bending tests shown here seem to be 3-point bending. It is acceptable that the authors follow the standards but it is worth considering that in 3-point bending the stress state is not a purely axial one but a complex axial+shearing stress state develops and the validity of the bending stress results is somewhat compromised . (The reviewer conducts 4-point bending for own test for this reason.)
    1. This statement regarding this issue has been declared on lines 359 to 364 and properly reference as the motive to display the modulus of elasticity from the parallel compression instead of the bending.
  7. Another issue is that equations 6 and 7 seem to be identical. Please correct.
    1. The equation 7 has been corrected regarding the utilization of the maximum load Fm as stated in line 229 and equation 6 with the load on the proportional limit in line 225
  8. Also, the modulus of elasticity is computed here in addition to the parallel compression section. Which one is shown in the tables, from compression tests or from bending?
    1. Compression, the motives are displayed in lines 359 to 364
  9. The physical testing and evaluation can be an important contribution if it involves new species not investigated previously, therefore it would be important to correct and amend this part of the manuscript as mentioned above.
    1. Has been added to conclusions in lines 492 to 494
  10. Entropy weighted COPRAS method.
    I have some concerns regarding this method. The discription of the method is lacking. At the start, in equations 13 and 16 terms xij are not defined. The major problem is that it is completely missing what input data the user should define and on what basis? Is it the xij values and how are they determined? The method is supposed (as it seems) to rank the alternatives on some basis but nothing is shown about it.
    1. The MCDM uses the table of characteristics of every material displayed in table 1, considering it a matrix with i an j positions, this has been stated in line 282 and 303 to 304
  11. The authors write (L351) that "Since the criteria weights has been delivered by the objective entropy method, ..." Does the term "objective" refer to that it gives objective results without user-specified parameters, or does it refer to the objective of a structural design, or something else?
    1. An objective weighting method describes that the weight of each criteria has been calculated by mathematical means without a subjective intervention of the selector such as preferences driven by some other experiences. This has been clarify in lines 457 to 458.
  12. It is not shown how the entropy values in Table 2 are obtained. To me the physical criteria listed in Table 2 seem user-specific, i.e. they depend on the objectives of the design.
    1. This is a subjective criteria, the aim of the entropy method is to rely only on a objective weighting driven by the mathematical difference calculated. The explanation where table 2 comes from was added in lines 453 and 454
  13. For example, in bridge design bending strength and flexural modulus are important while thermal conductivity is irrelevant, or in apartment buildings the thermal conductivity has larger importance while mechanical properties also matter, etc. As the ranking is one of the authors' conclusions, these issues need to be clarified.
    1. Considering that timber can be used in different applications, the general ranking and weighting can not be driven by a subjective method, in this sense the objective method Entropy allows to make an assessment driven by mathematical means. This has been added to conclusions on lines 495 to 497
  14. Also, as it seems, the method is only applied in this work without any original contribution, or if any part is the auhors' development, it is not clear to me.
    1. The MCDM has different applications and combinations on different subjects os science, in this case, there is no important literature regarding the utilization of the Entropy-COPRAS method for buildings. In this sense, this research used this approach as a contribution for a different material selection approach on this topic. This has been stated in lines 499 to 502
  15. : The conclusions on ranking heavily depend on the evaluation of the method as described above. As it is not shown how the results are obtained, it is also not known how sensitive the results are to the choice of the input values (whatever they are).
    1. The ranking relies on the mathematical method COPRAS which at the same time depends on the Entropy weighting metho. The Entropy-COPRAS method has been declared on the Method section, how and why it is used as stated before
  16. Moreover, the authors write that the results for Guayacan can be the consequence of its low moisture content. In fact, all investigated species have quite different moisture content, which makes the comparison less reliable. The authors may consider how the moisture content would affect the mechanical properties and thus the ranking as well. Therefore, these conclusions can be considered only as a starting point for further investigations, but definitely not exhaustive as they are. However, numerical results from the experiments have value and can be important.
    1. Having an awareness of the importance of the moisture content, the research was developed by using samples obtained from artisan production, hence it has not being altered but should be considered. On lines 133 to 137 it has been stated that the moisture efect has not been determinate since the characterization considers the state of the material at where they were acquired from the artisan producers, and in lines 437 to 439 it has been highlighted that this is an anormal variation that could be related to the artisan methods that should be improved. Also, a recommendation of this considerations has been added in lines 503 to 506

      Minor issues
  17. The use of technical terms is often wrong, inconsistent or inaccurate. The word "module" appears several time (L183, L272, etc), supposedly meaning "modulus" but "module" has a completely different meaning in English. I recommend to use the term "modulus of elasticity" throughout the text.
    1. Correction for the modulus of elasticity has been done on lines 9;12; 14; 231; 232; 450; 472; 473 and 476
  18. The authors write the term "test tube" a few times (L156,L175) but I have found no reference to it in the standard; if it is a reference to the test specimens, it is wierd as they are rectangular.
    1. The term test tube was wrong and was corrected for sample at lines 181 and 222
  19. The term "deformation" is used (L184) for d1, which is clearly not a deformation. I recommend the term "deflection" or "displacement".
    1. Ther term has been corrected on line 232
  20. The term "summatory" (L244) is an adjective, so use the noun "summation".
    1. The term has been corrected on line 318
  21. The term "charge" (L283) seems odd, mistyped maybe?
    1. The term charge was mistyped it was changed to load on line 380
  22. The figures display values for independent entities (wood species), so connecting the dots with solid lines makes no sense. It is advised to use bar plots or similar for such data.
    1. Figures has been changes to display data in bar plot
  23. Mistyped value in Table A4, column E, last row. Std should be 7.621.
    1. The error has been corrected on Table A5
  24. Clarity of some sentences is problematic. Sentence in L74-81 is too long and difficult to understand. The same for L90-94. Sentence in L287-288 has no verb in it. Etc.
    1. Corrections has been made in lines 91-95;108 to 113 and 384 to 385
  25. Referencing is not uniform. Use [number] consistently. Remove Name(year) in L90, L94-95, L97, and keep [number].
    1. The year in referencing has been removed leaving the authors name and the numbering only
  26. Provide the Latin names for the wood species investigated in this work. I can see the Guayacan in L58 but missing the Oak.
    1. Latin name has been added in line 60
  27. Use capital initials (F,T) for figures and tables in several places in the text, e.g. ...shown in Figure X.
    1. Capital letters has been added all over the manuscript
  28. Label G2P5 (L275) seems to be mistyped.
    1. Label has been corrected on line 372
  29. The use of English has a lot of minor errors, some typos but mostly grammatical errors. It includes inconsistent subject-verb matching, wrong word order, missing terms, etc. A thorough check and corrections are necessary.
    1. A proofreading has been made all over the text

We hope we have answered to all your observations, and we are thankful for the opportunity to improve

Best regards

Reviewer 2 Report

Comments and Suggestions for Authors

Dear authors,

The work is good, but to move on to the next step you must answer the following requirements.

1.      I think that the names of the faculties and universities belonging to the authors should be written in English.

2.      Lines 3-4. “…the Guayacan timber and the Ecuadorian Oak has not”. Please change in “…the Guayacan and the Ecuadorian Oak timber has not…”

3.      Line 12. A general conclusion on the work should exist at the end of the abstract.

4.      Line 31. “...energy efficiency”. A better expression would be: “…energetic efficiency”

5.      Line 34. It must be stated that the contradiction between the environmental part (preserving forests) and the industrial-productive part (industrial consumption of wood) is resolved at the level of the government policy of each country.

6.      Lines 39-43. The local name of the listed species does not tell us anything, no one outside the area will recognize these species. Please use the scientific-botanical name of these species, for a better understanding.

7.      Line 40. Complete ..."by their different colour."

8.      Line 52. Please change the word “woods” (keep on forestry) with “wood” or “timber”.

9.      Line 54. Please change the word “vaporized”.

10.   Line 58. “Caesalpinia paraguariensis”. A scientific name of wood specie is always written in italics: Caesalpinia paraguariensis.

11.   Lines 66-68. All standards must be found in references and cite them accordingly.

12.   Line 95. “…22]. Used…” Please change the point in comma.

13.   Line 101-107. For a better understanding of the content, it would be good to transform this long sentence (by which you express the objectives of the work) into two or three shorter ones.

14.   Lines 111-112. You must clearly specify why the density of the wood at 12% moisture content was not determined, this being known as the most important physical property of wood.

15.   Line 147. “…that 5 samples from every location were used”. Which standard stated that 5 samples are enough?

16.   Line 156. “…test tube.”. Change it in one of : sample/specimen/test piece, etc

17.   Lines 160-167. Great attention. The usual perpendicular compression is the one at the limit of proportionality between effort and deformation (only mentioned), whose methodology is completely different from what you present. This is because the test piece does not break during this test and is continuously compressed up to very high force values. Please give more explanations.

18.   Line 197. “…water free and constitution water” Please use the expressions: free water and bound water. Explanations are required.

19.   Lines 197-209. Specified the standard of moisture content determination (at beginning, not in the last rows).

20.   Line 209. Please use the sub-scripted value.

21.   Line 269. Figure 1 (and others). Please use the same FONT as is used in the text.

22.   Line 286. Please replace the “humidity” word, with “moisture content”.

23.   Line 331. Figure 3. Such a large variation in moisture content is not normal. This means that the method of obtaining moisture content is not the best. All wood properties are determined at 12% moisture content.

24.   Line 339. “weighting”. Please use another expression as: evaluating, grading, etc.

25.   Line 366. The chapter “Discussions” is completely missing. Please fill it out. Moreover, not too many comparisons with other authors can be observed in the results chapters. Only with standards.

26.   In the "Conclusions" chapter there are several conclusions, but they are not clearly highlighted through different paragraphs. Also, these conclusions must be simplified to be more visible.

27.   Lines 391-393. Some expressions must be erased.

28.   Lines 411-490. References. Some references have link of publishing and others not (Examples of number 6 and 10). Please unify all references from this point of view. Some references use the abbreviated form of journal, and others do not. Example: reference number 16 and 17.  Please unify all references from this point of view. 

Comments on the Quality of English Language

From a linguistic point of view (English) you have small lapses, these can be solved if someone carefully looks over the work. Some of the linguistic problems are highlighted in the review to the authors. Please take into consideration that the 28 aspects and requirements formulated to the authors are a necessary minimum that the authors must take into account.

Author Response

Dear reviewer,

Thank you for the time and work spent in the revision of our paper, it is very important for us to be considered for publication.

Regarding your observations.

  1. I think that the names of the faculties and universities belonging to the authors should be written in English.
    1. Our institutions have strict policies in how our papers should be cited. We can not change them
  2. Lines 3-4. “…the Guayacan timber and the Ecuadorian Oak has not”. Please change in “…the Guayacan and the Ecuadorian Oak timber has not…”
    1. Lines 3 and 4 has been corrected
  3. Line 12. A general conclusion on the work should exist at the end of the abstract.
    1. A general conclusion has been added from lines 12 to 16
  4. Line 31. “...energy efficiency”. A better expression would be: “…energetic efficiency”
    1. The expression has been corrected on line 36
  5. Line 34. It must be stated that the contradiction between the environmental part (preserving forests) and the industrial-productive part (industrial consumption of wood) is resolved at the level of the government policy of each country.
    1. The promotion of each government for a sustainable forest industry has been declared in lines 39 to 43,
  6. Lines 39-43. The local name of the listed species does not tell us anything, no one outside the area will recognize these species. Please use the scientific-botanical name of these species, for a better understanding.
    1. Scientific-botanical names has been added from lines 48 to 57
  7. Line 40. Complete ..."by their different colour."
    1. Colorado is a specie of tree, this line does not has any reference to coloration
  8. Line 52. Please change the word “woods” (keep on forestry) with “wood” or “timber”.
    1. Word has been changed on line 67
  9. Line 54. Please change the word “vaporized”.
    1. Word has been changed on line 70
  10. Line 58. “Caesalpinia paraguariensis”. A scientific name of wood specie is always written in italics: Caesalpinia paraguariensis.
    1. Style changed on line 74
  11. Lines 66-68. All standards must be found in references and cite them accordingly.
    1. Citation has been added in lines 82 to 84
  12. Line 95. “…22]. Used…” Please change the point in comma.
    1. Symbol changed on line 113
  13. Line 101-107. For a better understanding of the content, it would be good to transform this long sentence (by which you express the objectives of the work) into two or three shorter ones.
    1. The objective of the research has been separated in shorter paragraphs form lines 119 to 128
  14. Lines 111-112. You must clearly specify why the density of the wood at 12% moisture content was not determined, this being known as the most important physical property of wood.
    1. On lines 133 to 137 it has been stated that the density of the woods at 12% moisture has not been determinate since the characterization considers the state of the material at where they were acquired from the artisan producers.
  15. Line 147. “…that 5 samples from every location were used”. Which standard stated that 5 samples are enough?
    1. The standard does not limits the number of samples, however, 5 samples were used considered the limitations on the utilization of the equipment for all the tests. This has been stated on lines 180 to 181.
  16. Line 156. “…test tube.”. Change it in one of : sample/specimen/test piece, etc
    1. Word has been changed on line 181
  17. Lines 160-167. Great attention. The usual perpendicular compression is the one at the limit of proportionality between effort and deformation (only mentioned), whose methodology is completely different from what you present. This is because the test piece does not break during this test and is continuously compressed up to very high force values. Please give more explanations.
    1. The test is performed until a maximum penetration of 2.5mm considering the failure by crushing, This has been stated in lines 207 and 208. Also figure 5 has been added to illustrate the test.
  18. Line 197. “…water free and constitution water” Please use the expressions: free water and bound water. Explanations are required.
    1. A correction and better explanation has been added in line 251 to 253
  19. Lines 197-209. Specified the standard of moisture content determination (at beginning, not in the last rows).
    1. The specified standard has been moved to the beginning at lines 231 to 253
  20. Line 209. Please use the sub-scripted value.
    1. The sub-scripted values has been added at lines 265 and 266
  21. Line 269. Figure 1 (and others). Please use the same FONT as is used in the text.
    1. The font of the figures has been changed to match the font of the text
  22. Line 286. Please replace the “humidity” word, with “moisture content”.
    1. Word changed in line 382
  23. Line 331. Figure 3. Such a large variation in moisture content is not normal. This means that the method of obtaining moisture content is not the best. All wood properties are determined at 12% moisture content.
    1. In lines 436 to 438 it has been highlighted that this is an anormal variation that could be related to the artisan methods that should be improved.
  24. Line 339. “weighting”. Please use another expression as: evaluating, grading, etc.
    1. Regarding the multicriteria decision methods, it is widely used on the literature the term “weight” of the grade that the criteria receives, considering that it has a “weight“ on the determination of the best material for all the different MCDM. In this sense, we consider important to maintain the expression “weighting”
  25. Line 366. The chapter “Discussions” is completely missing. Please fill it out. Moreover, not too many comparisons with other authors can be observed in the results chapters. Only with standards.
    1. The present research has used the format of Results and discussion that has been corrected on line 336, the comparatives to other authors is limited to the scopes of the research, however, these can be found in lines 402 to 415; 431 to 435 ; 441 to 451 and 461 to 462
  26. In the "Conclusions" chapter there are several conclusions, but they are not clearly highlighted through different paragraphs. Also, these conclusions must be simplified to be more visible.
    1. The conclusions have been dived on different paragraph to be highlighted from lines 478 to 502
  27. Lines 391-393. Some expressions must be erased.
    1. The expressions has been erased
  28. Lines 411-490. References. Some references have link of publishing and others not (Examples of number 6 and 10). Please unify all references from this point of view. Some references use the abbreviated form of journal, and others do not. Example: reference number 16 and 17.  Please unify all references from this point of view.
    1. The reference has been unified

We hope we have answered to all your observations, and we are thankful for the opportunity to improve

Best regards

Round 2

Reviewer 1 Report

Comments and Suggestions for Authors

I accept the authors' answers.